# Solving text clustering problem using a memetic differential evolution algorithm

**Hossam M. J. Mustafa**[1] *, **Masri Ayob**[1], **Dheeb Albashish**[2], **Sawsan Abu-Taleb**[2]

**1** Data Mining and Optimization Research Group, Center of Artificial Intelligence Technology, Faculty of Information Science and Technology, University Kebangsaan Malaysia, Bangi, Malaysia, **2** Computer Science Department, Prince Abdullah bin Ghazi Faculty of Information and Communication Technology, Al-Balqa Applied University, Salt, Jordan

☯ These authors contributed equally to this work.
\* hasa.mustafa@gmail.com

**Data Availability Statement:** Data are available from the Laboratory of Computational Intelligence (LABIC) (http://sites.labic.icmc.usp.br/text_collections/ or http://sites.labic.icmc.usp.br/ragero/arffs/).The following are direct links to the datasets

## Abstract

The text clustering is considered as one of the most effective text document analysis methods, which is applied to cluster documents as a consequence of the expanded big data and online information. Based on the review of the related work of the text clustering algorithms, these algorithms achieved reasonable clustering results for some datasets, while they failed on a wide variety of benchmark datasets. Furthermore, the performance of these algorithms was not robust due to the inefficient balance between the exploitation and exploration capabilities of the clustering algorithm. Accordingly, this research proposes a Memetic Differential Evolution algorithm (MDETC) to solve the text clustering problem, which aims to address the effect of the hybridization between the differential evolution (DE) mutation strategy with the memetic algorithm (MA). This hybridization intends to enhance the quality of text clustering and improve the exploitation and exploration capabilities of the algorithm. Our experimental results based on six standard text clustering benchmark datasets (i.e. the Laboratory of Computational Intelligence (LABIC)) have shown that the MDETC algorithm outperformed other compared clustering algorithms based on AUC metric, F-measure, and the statistical analysis. Furthermore, the MDETC is compared with the state of art text clustering algorithms and obtained almost the best results for the standard benchmark datasets.

## Introduction

Data clustering is a common data mining task that has been applied in several applications to understand the hidden structures in data. It is considered as an essential task in several disciplines such as Information Retrieval [1], Internet of Things [2], Image segmentation [3], and wireless sensor networks [4]. Moreover, one of the widespread applications of data clustering is text clustering (TC), which is considered as an unsupervised learning method that operates without the prior knowledge of the text document labels [5]. The text clustering is utilized to cluster a vast quantity of disordered text documents as a result of the expanded big data and online information [6,7]. Thus, the text clustering aims to group a collection of text documents

used in our paper: OH15: http://sites.labic.icmc.
usp.br/text_collections/oh15.arff.zip; TR11: http://
sites.labic.icmc.usp.br/text_collections/tr11.arff.
zip; TR12: http://sites.labic.icmc.usp.br/text_
collections/tr12.arff.zip; TR41: http://sites.labic.
icmc.usp.br/text_collections/tr41.arff.zip; TR23:
http://sites.labic.icmc.usp.br/text_collections/tr23.
arff.zip; CSTR: http://sites.labic.icmc.usp.br/text_
collections/CSTR.arff.zip.

**Funding:** This work was supported by the FRGS of
Ministry of Higher Education, Malaysia. Ref. No:
FRGS/1/2018/ICT02/UKM/01/1.

**Competing interests:** The authors have declared
that no competing interests exist.

**Abbreviations:** ABC, Artificial bee colony; ACO, Ant
colony optimization; CS, Cuckoo search; DE,
Differential Evolution algorithm; FA, irefly
algorithm; GA, Genetic Algorithm; HS, Harmony
search algorithm; KHA, Krill herd algorithm; MA,
Memetic Algorithm; MDETC, Memetic differential
evolution for solving text clustering problems;
MMKHA, Hybrid krill herd algorithm; PSO, Particle
Swarm Optimizer; TC, Text clustering; TD, Text
document; C, Set of all Clusters; Cr, Crossover
constant; D, Set of all text documents; d, Text
document; F, DE Differentiation constant; i, Text
document number $i$; K, Number of all clusters;
MGI, Max generation without improvement; n,
Number of all text documents in set $D$; nl, Number
of documents in cluster l; Pool_Size,
Recombination mating pool size; Pop_Size, The
population size constant; t, Number of unique
terms that exist in the entire documents;
Tour_Size, Tournament selection size; wi,j, Weight
of term j in document $i$; Zl, The centroid of cluster $l$;
ABC, Artificial bee colony; ACO, Ant colony
optimization; CS, Cuckoo search; DE, Differential
Evolution algorithm; FA, Firefly algorithm; GA,
Genetic Algorithm; HS, Harmony search algorithm;
KHA, Krill herd algorithm; MA, Memetic Algorithm;
MDETC, Memetic differential evolution for solving
text clustering problems; MMKHA, Hybrid krill herd
algorithm; PSO, Particle Swarm Optimizer; TC, Text
clustering; TD, Text document; C, Set of all
Clusters; D, Set of all text documents; d, Text
document; F, DE Differentiation constant; i, Text
document number i; K, Number of all clusters; MGI,
Max generation without improvement; n, Number of
all text documents in set D; nl, Number of
documents in cluster l; Pool_Size, Recombination
mating pool size; Pop_Size, The population size
constant; t, Number of unique terms that exist in
the entire documents; Tour_Size, Tournament
selection size; wi,j, Weight of term j in document i;
Zl, The centroid of cluster l.

into a group of clusters according to the related contents and topics. A particular cluster may
include all related documents, and other clusters include irrelevant documents [8–10].

Recently, many researchers used the metaheuristic algorithms to address the text clustering
problem [6,8], such as krill herd algorithm (KHA) [10], particle swarm optimization (PSO)
[11]. The trade-off between exploration and exploitation in these algorithms plays a vital role
in improving the performance of the clustering algorithm, which can be enhanced to seek rea-
sonable clustering solutions based on specific datasets [12,13]. However, some algorithms
were unable to find robust and effective results across many datasets [10]. This may occur due
to an inefficient balance between exploitation and exploration that may lead to stagnation or
premature convergence [14]. Some recent studies have suggested hybridizing a local search
and a global search to obtain a good balance. The local search manages the exploitation,
whereas the global search manages the exploration [10,15–17].

Moreover, the optimization framework of the Memetic Algorithms (MAs) can utilize the
strength of different optimization algorithms by hybridizing them within the MA framework,
which may offer better performance [18]. The MA includes several evolutionary steps that
help in solving many complex optimization problems [18–21]. Consequently, the MA can be
hybridized with the Differential Evolution algorithm (DE), which revealed good performance
over several optimization problems.

Therefore, in this work, we propose a memetic differential evolution algorithm to address
the text clustering. The offered MDETC algorithm employs the approach of hybridization
between the Differential Evolution and the Memetic algorithms to address the text clustering
problem. The purposed text clustering algorithm aims to produce high-quality clustering mea-
sures such as AUC metric and F-measure.

## Related work

The primary task of text clustering is to group sets of documents into homogeneous clusters
[28]. This task can be achieved by employing a suitable similarity function that should be max-
imized/minimized the similarity between the documents clusters [6]. Several researchers have
used metaheuristic optimization algorithms to solve the text clustering problem such as
Genetic Algorithm [22,23], Particle Swarm Optimizer algorithm [24,25], Cuckoo search [26],
Ant colony optimization [27], Artificial bee colony algorithm [28,29], Firefly algorithm [30],
Harmony Search [31], and the hybrid metaheuristic approaches [32–37].

Some studies employed the Genetic Algorithm to address the text clustering problem. For
example, [22] proposed a text clustering method based on Genetic Algorithm that employed
the ontology and the thesaurus using several similarity measures. The researchers in [23] intro-
duced a text clustering method based on Genetic Algorithm, which was utilized separately to
every cluster to avoid the local optima. Moreover, the Particle Swarm Optimizer algorithm
used in some studies to reach an optimal solution. For example, [24] offered a hybridized Par-
ticle Swarm Optimizer with a k-mean algorithm with benchmark text datasets. The authors of
[25] proposed text clustering based on the Particle Swarm Optimizer algorithm (EPSO). Their
algorithm seeks a multi-local optimal solution using the Particle Swarm Optimizer.

Additionally, the researchers used the Cuckoo search to address the text clustering problem.
For example, [26] introduced a data clustering method based on and fuzzy cuckoo optimiza-
tion algorithm (FCOA) and the cuckoo optimization algorithm (COA). Other metaheuristics
such as Ant colony optimization utilized to solve the text clustering problem, for example, [27]
proposed a document clustering algorithm based on Ant colony optimization. Thus, the Artifi-
cial bee colony algorithm utilized to improve the text document clustering algorithm, for
example, [28] employed the chaotic map model in the local search to improve the exploitation

capability of the Artificial bee colony. The study of [29] utilized the Artificial bee colony algorithm in the text document clustering using the gradient search and the chaotic local search to enhance the exploitation capability of the Artificial bee colony.

Moreover, the Firefly algorithm (FA) used in [30] to address dynamic text document clustering using a Gravity Firefly Clustering (GF-CLUST). Other studies utilized the Harmony Search for the text document clustering. For example, [31] introduced the factorization approach to enhance the text document clustering.

The hybrid metaheuristic approaches are used to address the text clustering problem. For example, [32] combined Particle Swarm Optimizer with the Genetic Algorithm to address the text clustering problem. Their algorithm employed a Genetic Algorithm to enhance the global search and the Particle Swarm Optimizer to produce the range of search space. The researchers in [33] proposed a text clustering algorithm based on the combination between Particle Swarm Optimizer and Cuckoo search algorithm. Many other studies utilized the metaheuristic optimization algorithms to avoid the local optima problem of the K-means algorithm. For example, [34] applied the Harmony Search algorithm with text clustering to seek optimal clustering. Their proposed algorithm hybridized Harmony Search using the local search within the k-mean algorithm. The research of [35] hybridized k-mean with the Cuckoo search (CS) algorithm for addressing the web document clustering. The hybridization aims to improve the performance of web search results. The authors of [36] proposed a hybrid optimization algorithm to address the data clustering problem. Their algorithm combined the k-mean algorithm with the Tabu search (TS) to avoid the local optima problem. The researcher in [37] combined the Firefly algorithm with the k-mean algorithm. In their proposed algorithm, the Firefly algorithm employed to seek optimal centroids of the clusters that initialize the k-mean algorithm.

Other studies used the memetic differential evolution approach to solve several data clustering problems. For example, the study of [21] introduced a memetic differential evolution algorithm for solving data clustering problems. The algorithm proposes a clustering algorithm based on a modified adaptive Differential Evolution mutation algorithm and a local search algorithm to enhance the balance between exploration and exploitation. The experiments were based on several low dimensional real-life benchmark datasets obtained from the UCI repository of the machine learning databases. Additionally, the research employed the intra-cluster distance with the Euclidian distance similarity/dissimilarity function. Despite that this method was an effective approach to find reasonable clustering solutions, it may fail to find better solutions for high dimensional datasets such as text clustering problems. This may occur due to the utilization of inappropriate objective function that may lead to an imbalance between exploration and exploitation for high dimensional datasets.

Despite that the methods of the text clustering algorithms based on several metaheuristics approaches have better performance than other earlier algorithms, the problem of weak convergence exists in many metaheuristics algorithms. Specifically, the exploration and exploitation trade-off of the metaheuristics algorithms can be further enhanced.

## Contribution of this paper

This paper aims to tackle the issues discussed above, which can help in solving the text clustering problem by using a memetic differential evolution algorithm. More precisely, our contribution significance is two-fold.

1. We introduced a memetic Differential Evolution algorithm to address the text clustering problem. The introduced text clustering algorithm combined the MA and DE algorithms to solve the text clustering problem.

2. We developed a modified DE Mutation phase that can be applied to enhance the search of the text clustering algorithm.

More specifically, the proposed text clustering algorithm utilizes a DE mutation that is coupled with the memetic algorithm evolutionary steps. The mutation step intends to improve the search abilities of DE by employing an adaptive mutation strategy. Moreover, the improvement phase is modified to remove the duplicated solutions, which aim to avoid falling into premature convergence. The restart phase was modified by replacing a portion of the population with new solutions that are randomly generated to improve the diversity of the population.

### The organization of the paper

This paper contains the following sections: The second section presents the concepts and background such as text clustering, DE and MA. In the third section, discusses the proposed memetic DE for the text clustering problem. The fourth section discusses the results of the MDETC algorithm experiments. Lastly, the fifth section discusses the conclusions and future works of the research.

## Background

This section presents the necessary concepts of text clustering problem, memetic algorithm, and differential evolution (DE) algorithm, which are employed in the offered data text clustering algorithm.

### Text clustering problem

Text document clustering is a method of splitting a set of $n$ text documents into a group of $K$ clusters, which can be grouped using a particular dissimilarity/similarity measure. The $n$ text documents are denoted by a set $D = \{d1, d2, \ldots, dn\}$, the $K$ clusters are represented by $C = \{C1, C2, \ldots, CK\}$, where the entire text documents in each cluster are similar, and other text documents are dissimilar. Thus, the number of clusters is given in advance [10,38].

The pre-processing steps of the text should be used to decrease the number of text attributes/features to support the algorithm task. The pre-processing steps are organized into (a) Tokenization (b) Removal stop word (c) Stemming (d) Feature selection and (e) Calculate the terms weighing [10]. The text documents can be represented by the vector space model (*VSM*) as presented in Eq (1). *VSM* model denotes each document $i$ as a vector of length $t$ [10].

$$VSM = \begin{bmatrix} w_{1,1} & w_{1,2} & \cdots & w_{1,(t-1)} & w_{1,(t)} \\ \cdots & \cdots & \cdots & \cdots & \cdots \\ \vdots & \vdots & w_{i,j} & \vdots & \vdots \\ w_{(n-1),1} & w_{(n-1),2} & \cdots & \cdots & w_{(n-1),t} \\ w_{n,1} & w_{n,2} & \cdots & w_{n,(t-1)} & w_{n,t} \end{bmatrix} \tag{1}$$

The $wi,j$ denotes the value of the *tf/idf* weight of term $j$ in document $I$, which is commonly used term weighting method that measures whether the term is frequent or rare across all documents [24], and calculated using Eq (2). The *tf (i,j)* denotes the frequency of term $j$ in document $i$, and $n$ denotes the total number of documents in $D$, the *df (j)* is term $j$ frequency in all documents [10]:

$$w_{i,j} = tf(i,j) \times \log\left(\frac{n}{df(j)}\right) \tag{2}$$

The text document clustering problem can be formulated in Eq (3):

$$\underset{C}{Optimize}\ f(D, C) \tag{3}$$

The *f(D, C)* represents the fitness function that measures the quality of the clusters that is produced by the text clustering methods. Hence, the fitness function can be minimized or maximized subject to the employed dissimilarity/similarity measure. The quality of the text clustering solutions can be measured by the intra-cluster distance dissimilarity/similarity measure, which is commonly utilized in text clustering [10], as shown in the Eq (4):

$$f(D, C) = \sum_{l=1}^{k} \sum_{di \in Cl}^{n} d(d_i, Z_l) \tag{4}$$

The *d(di, Zl)* denotes the distance between the centroid of cluster *Zl* and text document *di*. The cosine distance is one of the most widely used distance functions in text clustering [10,38]. It can measure the similarity between document *di* and the centroid of cluster *Zl* inside the same cluster, as in Eq (5).

$$d_{\text{cosine}}(d_i, Z_l) = \frac{\sum_{j=1}^{t} w(d_i, t_j) \times w(Z_l, t_j)}{\sqrt{\sum_{j=1}^{t} w(d_i, t_j)^2} \sqrt{\sum_{j=1}^{t} w(Z_l, t_j)^2}} \tag{5}$$

The *w(Zl, tj)* denotes the weight of term *j* in the centroid number *l, and w(di, tj)* denotes the weight of term *j* in document *i*. Additionally, centroids *Zl* can be manipulated as the average value of the entire cluster text documents, as shown in Eq (6). The *nl* denotes the number of text documents in cluster *Zl*.

$$Z_l = \frac{1}{n_l} \sum_{\forall O_i \in Z_l} (d_i) \tag{6}$$

## Differential evolution algorithm

The Differential Evolution algorithm (DE) is considered as an effective metaheuristic evolutionary algorithm that was introduced to solve continuous and combinatorial optimization problems [19]. DE begins by population initialization. At every iteration, parents are chosen from solutions for the crossover and mutation, to produce the trial solution [19]. The mutation phase is responsible for perturbing the solution by a scaled differential vector, which includes many randomly chosen solutions to generate the mutant solution. The parent solutions are compared with the offspring solution utilizing the fitness function; the better one is then selected as the new solution to the subsequent iteration. The algorithm terminates when a condition is met, and the problem's solution is chosen as the best individual in the population.

## Memetic algorithms

Memetic Algorithm (MA) is a metaheuristic algorithm that combines the problem-specific solvers with the evolutionary algorithm. The solvers can be performed as an approximation, local or exact search heuristics. The combination intends to find better solutions and find unreachable solutions by the local search methods or the evolutionary algorithms alone. Besides, MAs provide an optimization framework that integrates various local search strategies, learning strategies [39], perturbation mechanisms, and population management strategies [40]. MAs have several names in the literature, such as Lamarckian EA, hybrid Genetic Algorithm, or Baldwinian evolutionary algorithm.

| $d_1$ | $d_2$ | $d_3$ | $d_4$ | $d_5$ | $d_6$ | $d_7$ | $d_8$ | $d_9$ |
|-------|-------|-------|-------|-------|-------|-------|-------|-------|
| C1 | C1 | C2 | C2 | C1 | C2 | C1 | C1 | C2 |

**Fig 1. Example of the label-based representation of a candidate solution.**

The MA utilized other optimization algorithms by employing them inside the framework [41]. For example, metaheuristic algorithms such as Differential Evolution has shown better mutation performance [42] with appropriate parameter settings and mutation strategies. The combination of DE within the MA can offer three benefits: Firstly, the offspring's quality produced by evolutionary algorithms such as MA can be improved by implementing several search methods in the optimization search process. An example of these search methods is the DE mutation, which can be employed to generate better quality individuals [43]. Secondly, premature convergence and stagnation can be minimized when employing a DE algorithm by balancing exploitation and exploration, which can be achieved by utilizing several mutation strategies [19]. Thirdly, the DE population can stagnate when the offspring are less fit than their parents over a given number of iterations. To address this, the DE's performance can be enhanced by employing a convenient hybridization including local search algorithms within the MA framework [43,44].

The Memetic Algorithm includes the initialization procedure that creates solutions of the initial population; the compete procedure that is utilized to reconstruct the current population using the previous population, and the restart procedure, which is started on every degenerate state of the population [45].

## Proposed algorithm

This section describes the evolutionary steps and the solution representation of the introduced MDETC algorithm.

## Solution representation

The label-based solution representation is employed to represents the candidate solution in the text clustering problem. Each solution represents a set of $n$ documents that contain the cluster number related to each document. Fig 1 shows an example of the label-based representation of a candidate solution that contains two clusters and nine documents.

Moreover, a centroid-based 2-dimensional array is employed in the local search to store the centroid values of the clusters. The array includes $D$ columns and $K$ rows, where the total number of the attributes is denoted by $D$, and $K$ is the number of the clusters. Fig 2 presents an example of a candidate solution of a dataset that contains two attributes and two clusters.

|  | Attribute 1 | Attribute 2 |
|-----------|-------------|-------------|
| Cluster 1 | 4.3 | 5.3 |
| Cluster 2 | 4.5 | 3.4 |

**Fig 2. Example of the centroid-based representation of a candidate solution.**

## The MDETC proposed algorithm

In MDETC, the DE mutation is hybridized with the evolutionary steps of the MA that utilizes an adaptive strategy DE/current-to-best/1. The hybridization aims to improve the convergence rate. Thus, premature convergence can be prevented in the restart step by rebuilding the diversity of the population. At last, the improvement step plays an important role to seek better solutions. The pseudo-code for the proposed MDETC algorithm is presented in Fig 3, which consist of the following phases:

**The population initialization phase.** The initial solutions of MDETC are randomly generated. The documents are grouped into $K$ random clusters; every cluster's centroid is computed by Eq (6). These steps are repeated to produce $Pop\_Size$ random solutions.

---

The MDETC algorithm

---

1:      $Max\_Itr$ = maximum number of iterations
        $MGI$ = max generation without improvement
        $Pop\_Size$ = the population size
        $Pool\_Size$ = mating pool size
        $Tour\_Size$ = Tournament selection size
        $Cr$ = crossover constant
        $F$ = differentiation constant
        create an empty population $population$
        $imp\_idex = 0$
        **for i=1 : PopSize**
2:        randomly initialize solutions in $population$
3:      **end for**
4:      Sort $population$
5:      $Best\_solution$ = population(1)
6:      **for i=1 : Max\_Itr**
7:       $population$ = Recombination()
8:       $population$ = DE\_Mutation()
9:       $population$ = Improve\_population()
10:      Sort $population$
11:      **if** $population$(1) is better than Best\_solution **then**
12:            $Best\_solution$ = $population$(1)
13:            $imp\_idex = 0$
14:        **else**
15:            $imp\_idex = imp\_idex + 1$
16:      **endif**
17:       **if** $imp\_idex$ >= MGI **then**
18:            $population$ = Restart \_Population()
19:            $imp\_idex = 0$
20:         **endif**
21:      **end for**
22:      **return** $Best\_solution$

---

**Fig 3. The pseudo-code of the proposed MDETC algorithm.**

---

Creating a trial solution

---

```
1:       create_Trial_Sol(s1, s2, s3, Curr_Iteration)
2:       begin
3:         Create an empty solutions Trail_sol
4:       for i=1 : NoClusters
5:         for j=1 : NoAttributes
6:           z1= centroid(s1,i,j)
7:           z2=  centroid(s2,i,j)
8:           z3=  centroid(s3,i,j)
9:           centroid(Trail_sol,i,j)=z1+
                        (z2 - z3) * (1 - Curr_Itr  / Max_Itr )
10:        end for
11:      end for
12:      find nearest cluster to documents in Trail_sol solution;
13:      calculate fitness value for Trail_sol solution;
14:      return Trail_sol;
15:      end
```

---

**Fig 4. The pseudo-code of creating a trial individual algorithm.**

**The recombination phase.**   The mating pool approach [46] is employed in this phase with a size of *Pool_Size*. This phase also employs the tournament selection with a size of *Tour_Size* [47], which is combined with the mating pool. The two-point crossover is then applied to the mating pool. At last, the population is joined with the mating pool, where the worst individuals in the population are replaced with new individuals from the mating pool.

**The DE mutation phase.**   This phase utilizes the *DE/current-to-best/1* strategy [21], as shown in Fig 3. The cluster centroids are adjusted in the mutation step to obtain better solutions, as presented in Fig 4. This is accomplished with Eq (7). The *Zbest* is the best solution centroid, *Zi* denotes the current solution centroid, *Zrand* denotes a random centroid, the *Curr_Iteration* denotes the current MDETC algorithm iteration number, and *Max_Iterations* is the maximum number of iterations of MDETC.

$$Z_i = Z_{current} + \left( (Z_{best} - Z_{rand}) \times \left( 1 - \frac{Curr\_Iteration}{Max\_Iterations} \right) \right) \tag{7}$$

**The improvement phase.**   The improvements step clears the duplicated solutions, which guarantees better diversity in the population to prevent any premature convergence.

**The restart phase.**   Whenever the population falls into the degeneration state, the restart step is invoked [45]. The restart strategy retains some portion of the population and excepts the other solutions by generating new solutions. The MDETC preserve 75% of the population for the subsequent iteration, while the rest of the population is produced randomly.

## Experimental results and setup

### Experimental setup

The MDETC performance is studied using six standard real datasets from the Laboratory of Computational Intelligence (LABIC) and represented in numerical form after the extraction of

---

**Table 1. The characteristics of the used LABIC datasets.**

| Dataset | Source | No. of documents | No. of terms | No. of clusters |
|---|---|---|---|---|
| CSTR | Technical Reports | 299 | 1725 | 4 |
| tr41 | TREC | 878 | 7454 | 10 |
| tr12 | TREC | 313 | 5804 | 8 |
| tr23 | TREC | 204 | 5832 | 6 |
| tr11 | TREC | 414 | 6429 | 9 |
| oh15 | MEDLINE | 913 | 3100 | 10 |

the terms. These datasets contain different variety of characteristics, such as the number of terms, clusters, and documents, and variety of complexity [48], where the datasets that been used are CSTR, tr41, tr23, tr12, tr11, and oh15, as shown in Table 1. To assess the efficiency of the introduced algorithm, the performance of MDETC is compared with the K-means algorithm, DE [21], and Genetic Algorithm (GA) [22], where the algorithms are implemented using the same experimental setup.

The algorithms' performance is evaluated using the F-measure, which matches the ground truth with the obtained clustering solution to identify the correspondence between them. Also, the receiver operating characteristic curves (ROC) are plotted and the area under the curve (AUC) metric was calculated. A higher value of the AUC metric and F-measure means better quality of the clustering algorithm, which both range from 0 to 1.

The ROC curve can measure the degree of separability, which shows the capability of the algorithm to distinguish between classes. The ROC curves are plotted using the True Positive Percentage (TPP) against the False Positive Percentage (FPP). The TPP and FPP are computed using Eq (8) and Eq (9).

$$\text{TPP} = \frac{\text{Number of true positives}}{\text{Number of true positives} + \text{Number of false negatives}} \tag{8}$$

$$\text{FPP} = \frac{\text{Number of false positives}}{\text{Number of true negatives} + \text{Number of false positives}} \tag{9}$$

The F-measure of cluster $S_j$ can be computed using the recall and precision, which are shown in Eq (10) and Eq (11), Where $Nij$ denoted the number of objects of class $Ci$ in cluster $Sj$, $|Sj|$ is the number of objects in cluster S$j$, and $|Ci|$ is the number of objects in class $Ci$. The F-measure is computed using Eq (12).

$$recall(C_i, S_j) = \frac{N_{ij}}{|C_i|} \tag{10}$$

$$precision(C_i, S_j) = \frac{N_{ij}}{|S_j|} \tag{11}$$

$$F - measure(C_i, S_j) = \frac{2 \times precision(C_i, S_j) \times recall(C_i, S_j)}{precision(C_i, S_j) + recall(C_i, S_j)} \tag{12}$$

The settings of the parameters of the MDETC algorithm were separately tested 31 times on all datasets; the average values of the AUC metric and F-measure were calculated. The parameter setting of the proposed MDETC is shown in Table 2, which is based on an experimental basis and the drawing on previous work from the scientific literature [21]. At last, the

**Table 2. Parameters setting used in experiments.**

| parameter | Value |
|---|---|
| No. of generations | 100 |
| Population size | 20 |
| Tournament selection size | 10 |
| Recombination mating pool size | 10 |
| Max Gen without improve | 20 |
| Crossover probability | 0.9 |
| DE mutation scaling factor | 0.7 |

algorithms are applied using Oracle Java 1.8, where it was run on a personal computer with an Intel Core i7 CPU (2.6GHz) and a RAM of 8 GB size.

## Experimental results and discussion

Table 3 shows the average results of the AUC metric obtained by MDETC and the competing algorithms. The proposed MDETC achieved the best results on tr23, tr12, tr41, CSTR, and oh15 datasets, also it achieved the second-best result on the tr11 dataset. Based on AUC metric results, the MDETC obtained an excellent performance on tr41, CSTR, and oh15 datasets. Besides, MDETC obtained fair performance on tr23, tr11, and tr12 datasets. The results show that the proposed MDETC algorithm has a higher AUC metric compared with the competing algorithms, for example, the results of tr41 dataset indicates that MDETC obtained an AUC metric value of 0.9511, whereas the F-measure results of K-means, DE, and GA are 0.5533, 0.5081, and 0.49, respectively.

Moreover, the results in Table 3 are further analyzed using the rankings generated by Friedman's test based on the AUC metric, as shown in Table 4. Friedman's test has shown that MDETC obtained a significant difference with a $p$-value of 0.03207 that is below the significance level ($\alpha = 0.05$). The results confirm that MDETC obtained the best ranking based on the AUC metric. The DE obtained the second-best rank, and then the GA algorithm. Finally, K-means achieved the worst rank.

Moreover, the statistical difference between the control case (MDETC) and the other algorithms is detected using the Holm's post-hoc procedure. Table 5 demonstrates the $p$-value achieved by Holm's procedure, where the null hypothesis is rejected based on the achieved $p$-value that needs to be less than the adjusted value of α ($\alpha/i$). The value of $i$ represents the rank of each algorithm. The Holm's procedure demonstrates that MDETC is statistically better than K-means, DE and GA based on the AUC metric.

Fig 5 shows the corresponding ROC curves obtained by the MDETC, K-means, DE and GA algorithms on the used datasets. The ROC curves demonstrate that MDETC produces

**Table 3. The comparison of AUC values obtained by the MDETC, K-means, DE and GA algorithms.**

| Dataset | K-means | DE | GA | MDETC |
|---|---|---|---|---|
| tr23 | 0.4697 | 0.5 | 0.4457 | **0.5575** |
| tr11 | 0.4745 | 0.4701 | **0.5212** | 0.5206 |
| tr12 | 0.4259 | 0.4438 | 0.4524 | **0.4577** |
| tr41 | 0.5533 | 0.5081 | 0.49 | **0.9511** |
| CSTR | 0.5555 | 0.5706 | 0.5337 | **0.802** |
| oh15 | 0.5335 | 0.5588 | 0.5635 | **0.9052** |

**Table 4. Friedman test ranking for MDETC, K-means, DE and GA algorithms based on the AUC metric.**

| Algorithm | Ranking |
|---|---|
| MDETC | 1.1666 |
| DE | 2.8333 |
| GA | 2.8333 |
| K-means | 3.1666 |

excellent performance on tr41, CSTR, and oh15 datasets with better capability to distinguish between classes. Thus, MDETC obtained fair performance compared with the competing algorithms on tr23, tr11, and tr12 datasets.

Table 6 demonstrates the average results of the F-measure obtained by competing algorithms. The proposed MDETC achieved the best results on all datasets concerning the F-measure (i.e., tr23, tr11, tr12, tr41, CSTR, and oh15). The results show that the proposed MDETC algorithm has a higher F-measure compared with the competing algorithms, for example, the results of CSTR dataset indicates that MDETC obtained an F-measure value of 0.6908, whereas the F-measure results of K-means, DE, and Genetic Algorithm (GA) are 0.5008, 0.5429, and 0.5133, respectively. However, the F-measure result achieved by GA is close to MDETC on tr12 datasets.

Moreover, the results in Table 6 are further analyzed using the rankings generated by Friedman's test based on the F-measure, as shown in Table 7. Friedman's test has shown that MDETC obtained a significant difference with a $p$-value of 0.00974 that is below the significance level ($\alpha = 0.05$). The results confirm that MDETC obtained the best ranking based on the F-measure. The DE obtained the second-best rank, and then the K-means algorithm. Finally, GA achieved the worst rank.

Moreover, the statistical difference between the control case (MDETC) and the other algorithms is detected using the Holm's post-hoc procedure. Table 8 demonstrates the $p$-value achieved by Holm's procedure, where the null hypothesis is rejected based on the achieved $p$-value that needs to be less than the adjusted value of $\alpha$ ($\alpha/i$). The Holm's procedure demonstrates that MDETC is statistically better than K-means, DE and GA.

Fig 6 shows the convergence curves on the employed datasets. The curves demonstrate that MDETC produces the best convergence performance on the six datasets with fast convergence in the initial iterations; next, convergence becomes slower. The proposed memetic steps improved efficiency by avoiding premature convergence. The DE obtained the second best convergence rate results, and GA obtained the worst results.

Table 9 shows the running time of a single iteration of the proposed MDETC, K-means, DE and GA algorithms on the related datasets to investigate the complexity of these algorithms. As presented in Table 9, The GA algorithm obtained the best results of the processing time on all datasets. Nevertheless, the MDETC requires less processing time compared to DE and K-means algorithms on the employed datasets except for the tr23 dataset. The K-means achieved the best third-best processing time on the entire datasets except for the tr12 dataset. The DE did not achieve any shorter running time on the test datasets except for the tr12 dataset.

**Table 5. Comparison between MDETC, K-means, DE and GA algorithms using Holm's post-hoc procedure based on the AUC metric.**

| $i$ | Algorithm | $\alpha/i$ | $p$-value of Holms | Null Hypothesis |
|---|---|---|---|---|
| 1 | DE | 0.05/1 = 0.0500 | 0.02534 | Rejected |
| 2 | GA | 0.05/2 = 0.0250 | 0.02434 | Rejected |
| 3 | K-means | 0.05/3 = 0.0166 | 0.00729 | Rejected |

Consequently, the trade-off between the time-cost and the quality problem appeared, where hybrid metaheuristic methods, such as MDETC, can achieve optimal solutions in acceptable running time. On the other hand, the traditional metaheuristic algorithm does not promise to

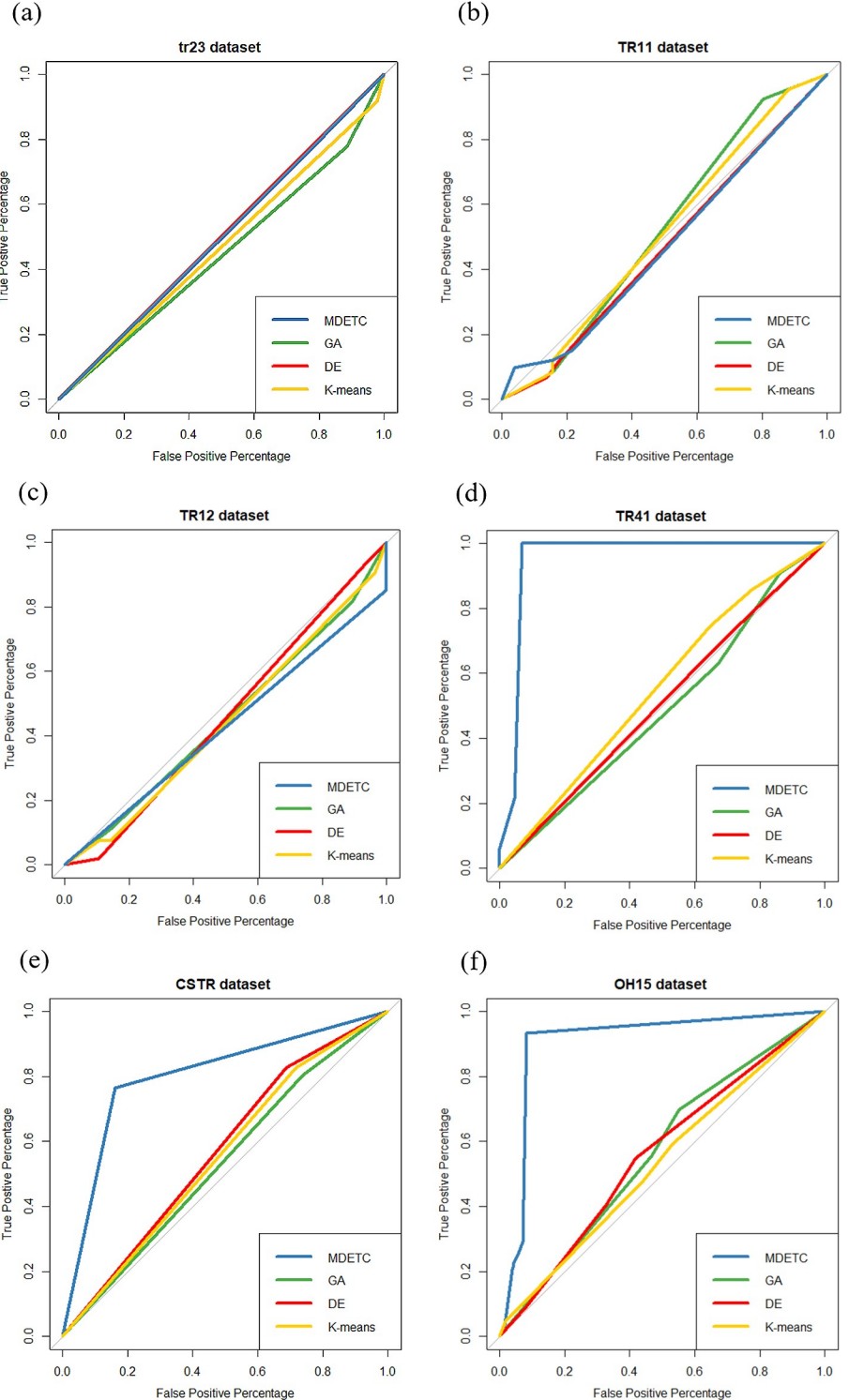

**Fig 5.** The ROC curves on (a) tr23, (b) tr11; (c) tr12; (d) tr41; (e) CSTR; (f) oh15 datasets.

**Table 6. The comparison of F-measure values obtained by the MDETC, K-means, DE and GA algorithms.**

| Dataset | K-means | DE | GA | MDETC |
|---------|---------|-----|-----|-------|
| tr23 | 0.5759 | 0.5791 | 0.5572 | **0.6240** |
| tr11 | 0.5043 | 0.4398 | 0.4595 | **0.5414** |
| tr12 | 0.3402 | 0.4114 | 0.4470 | **0.4481** |
| tr41 | 0.4494 | 0.4030 | 0.3685 | **0.6269** |
| CSTR | 0.5008 | 0.5429 | 0.5133 | **0.6908** |
| oh15 | 0.3709 | 0.2976 | 0.2788 | **0.5895** |

**Table 7. Friedman test ranking for MDETC, K-means, DE and GA algorithms based on the F-measure.**

| Algorithm | Ranking |
|-----------|---------|
| MDETC | 1 |
| DE | 2.833 |
| K-means | 2.833 |
| GA | 3.333 |

**Table 8. Comparison between MDETC, K-means, DE and GA algorithms using Holm's post-hoc procedure based on the F-measure.**

| i | Algorithm | α/i | p-value of Holms | Null Hypothesis |
|---|-----------|-----|------------------|-----------------|
| 1 | DE | 0.05/1 = 0.0500 | 0.013906 | Rejected |
| 2 | K-means | 0.05/2 = 0.0250 | 0.013906 | Rejected |
| 3 | GA | 0.05/3 = 0.0166 | 0.001745 | Rejected |

obtain the optimal solution and commonly can produce sub-optimal and good-quality solutions in shorter running time.

## Comparison between MDETC and state of the art

The performance of MDETC is compared with the state of the art algorithms, such as the hybrid krill herd algorithm (MMKHA) [10], krill herd algorithm (KH) [10], particle swarm optimization (PSO) [49], Hybrid Harmony Search (HS) [34]. As presented in Table 10, the F-measure achieved by MDETC is better than competing algorithms. The MDETC obtained the optimum F-measure on the tr23, tr11, tr41, CSTR, and oh15 datasets. The MMKHA algorithm obtained the optimum F-measure on the tr12 dataset and scored the second-best result on the remaining datasets. The results presented in Table 10 reveals that MDETC achieved consistent performance across all datasets using the F-measure.

Additionally, the results in Table 10 are further analyzed using the rankings generated by Friedman's test based on the F-measure, as shown in Table 11. The test has shown that MDETC obtained a significant difference with a p-value of 0.01302 that is below the significance level ($\alpha = 0.05$). The results confirm that MDETC obtained the best ranking based on the F-measure. The MMKHA algorithm attained the second-best rank, and the PSO scored the third rank, then the KH. Finally, HS achieved the worst rank. The rankings presented in Table 11 show that MDETC performance based on the F-measure is consistent when compared with the state of art algorithms.

Moreover, Table 12 shows the p-value of MDETC and the state of art algorithms using Holm's post-hoc procedure, where the null hypothesis is rejected based on the achieved p-

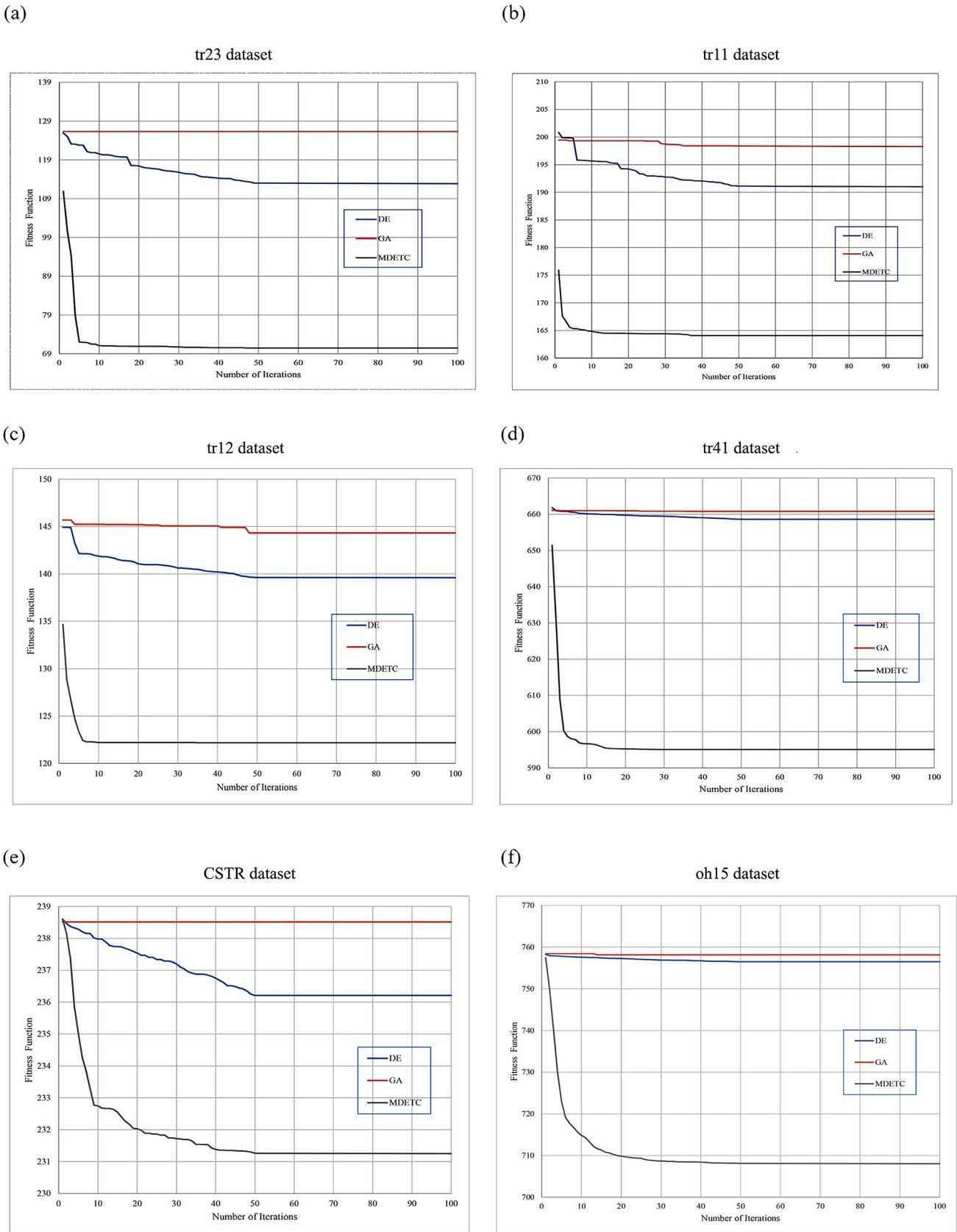

**Fig 6.** The convergence curves on (a) tr23, (b) tr11; (c) tr12; (d) tr41; (e) CSTR; (f) oh15 datasets.

**Table 9. Running time of MDETC, K-means, DE and GA algorithms.**

| Dataset | K-means | DE | GA | MDETC |
|---------|---------|-------|-------|-------|
| tr23 | 0.301 | 0.517 | **0.052** | 0.362 |
| tr11 | 1.001 | 1.211 | **0.070** | 0.831 |
| tr12 | 0.784 | 0.770 | **0.059** | 0.533 |
| tr41 | 1.211 | 2.907 | **0.192** | 2.002 |
| CSTR | 0.201 | 0.246 | **0.021** | 0.171 |
| oh15 | 1.109 | 1.343 | **0.069** | 0.918 |

**Table 10. F-measure comparison between MDETC and the state of art algorithms.**

| Dataset | HS | KH | PSO | MMKHA | MDETC |
|---------|--------|--------|--------|--------|---------|
| tr23 | 0.4021 | 0.4004 | 0.3565 | 0.4214 | **0.6240** |
| tr11 | 0.4095 | 0.4138 | 0.4380 | 0.5164 | **0.5414** |
| tr12 | 0.4526 | 0.5019 | 0.4708 | **0.5624** | 0.4481 |
| tr41 | 0.4392 | 0.4272 | 0.4471 | 0.5241 | **0.6269** |
| CSTR | 0.5268 | 0.4847 | 0.5090 | 0.6055 | **0.6908** |
| oh15 | 0.4185 | 0.4840 | 0.4471 | 0.5278 | **0.5895** |

**Table 11. Friedman test ranking for MDETC and the state of art algorithms based on the F-measure.**

| Algorithm | Ranking |
|-----------|---------|
| MDETC | 1.6666 |
| MMKHA | 1.8333 |
| PSO | 3.6666 |
| KH | 3.8333 |
| HS | 4.0 |

**Table 12. Comparison between MDETC and the state of art algorithms using Holm's procedure based on the F-measure.**

| i | Algorithm | α/i | p-value of Holms | Null Hypothesis |
|---|-----------|-----|------------------|-----------------|
| 1 | MMKHA | 0.05/1 = 0.0500 | 0.85513 | Not rejected |
| 2 | PSO | 0.05/2 = 0.0250 | 0.02445 | Rejected |
| 3 | KH | 0.05/3 = 0.0166 | 0.01762 | Rejected |
| 4 | HS | 0.05/4 = 0.0125 | 0.01058 | Rejected |

value that needs to be less than the adjusted value of α (α/i). The Holm's procedure shown in Table 12 demonstrates that MDETC is statistically better than PSO, KH, and HS. Thus, MDETC is not significantly different from the MMKHA algorithm. However, the results presented in Table 10 confirm that the MDETC algorithm outperformed the MMKHA based on the tested datasets.

## Conclusions and future work

This work proposed an MDETC algorithm for addressing the text clustering problem. The combination of DE and Memetic algorithms intends to achieve a better balance between exploration and exploitation. The algorithm introduced a DE mutation operator that is

hybridized within the Memetic algorithm. To prove the effectiveness of the introduced algorithm, six standard text clustering benchmark datasets (i.e. the Laboratory of Computational Intelligence (LABIC)) employed to assess the presented algorithm. The Experimental results confirmed that the introduced MDETC algorithm obtained consistent performance compared to the state of art algorithms concerning the AUC metric and F-measure validity measures. These results revealed that the proposed MDETC has achieved a better balance between exploration and exploitation and improved the performance of the Memetic algorithms to solve the text clustering problem. The MDETC algorithm obtained the optimum results of the F-measure on tr23 (62.4%), tr11 (54.14%), tr41 (62.69%), CSTR (69.08%), and oh15 (58.95%) datasets. Furthermore, the future work will concentrate on incorporating different validity measures when employed within the multi-objective metaheuristic algorithms.

## Supporting information

**S1 File.**
(DOCX)

## Author Contributions

**Funding acquisition:** Masri Ayob.

**Investigation:** Hossam M. J. Mustafa.

**Methodology:** Hossam M. J. Mustafa, Dheeb Albashish, Sawsan Abu-Taleb.

**Software:** Hossam M. J. Mustafa, Dheeb Albashish, Sawsan Abu-Taleb.

**Supervision:** Masri Ayob.

**Writing – original draft:** Hossam M. J. Mustafa.

**Writing – review & editing:** Masri Ayob.

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
