## [Decision Letter · Decision Letter 0]

25 Sep 2019

PONE-D-19-20564

Solving Text Clustering Problem using a Memetic Differential Evolution Algorithm

PLOS ONE

Dear Mr. Mustafa,

Thank you for submitting your manuscript to PLOS ONE. After careful consideration, we feel that it has merit but does not fully meet PLOS ONE’s publication criteria as it currently stands. Therefore, we invite you to submit a revised version of the manuscript that addresses the points raised during the review process.

The reviewers think that your article has some merits but needs to address their comments and suggestions in order to improve the quality of the article.

We would appreciate receiving your revised manuscript by Nov 09 2019 11:59PM. To enhance the reproducibility of your results, we recommend that if applicable you deposit your laboratory protocols in protocols.io, where a protocol can be assigned its own identifier (DOI) such that it can be cited independently in the future. For instructions see: http://journals.plos.org/plosone/s/submission-guidelines#loc-laboratory-protocols

We look forward to receiving your revised manuscript.

Kind regards,

Mohd Nadhir Ab Wahab, Ph.D.

Academic Editor

PLOS ONE

Journal Requirements:

Additional Editor Comments:

The reviewers think that your article has some merits but needs to address their comments and suggestions in order to improve the quality of the article.

Reviewers' comments:

Reviewer's Responses to Questions

**Comments to the Author**

1. Is the manuscript technically sound, and do the data support the conclusions?

Reviewer #1: Yes

Reviewer #2: Partly

2. Has the statistical analysis been performed appropriately and rigorously? 

Reviewer #1: No

Reviewer #2: No

3. Have the authors made all data underlying the findings in their manuscript fully available?

Reviewer #1: Yes

Reviewer #2: Yes

4. Is the manuscript presented in an intelligible fashion and written in standard English?

Reviewer #1: No

Reviewer #2: No

5. Review Comments to the Author

Reviewer #1: The introduction provides a good, generalized background of the topic that quickly gives the reader an appreciation of the wide range of applications for this technology. I think the motivations for this study need to be made clearer.Figures must be in a good quality with high resolution.The manuscript is poorly written. It is full of incomplete & meaningless sentences. I really find it difficult to understand the manuscript. Authors really need to improve their written communications skills in English. All references must be revised. Issue numbers and volume numbers are missed from most of references. Conclusions should address whether or not they achieved the objective of the study.

Reviewer #2: I congratulate the author on their work, however there seem to be multiple issues that need to be addressed before publication:

(1) First and foremost, there are multiple grammatical and editorial errors in the text. As a non-native English speaker I can understand the text, but a larger audience will have a problem with that. Also it speaks of lack of attention: For example in the introduction section, one to the last paragraph is an incomplete sentence/paragraph "The reseach [25] of we propose a memetic differential evolution algorithm to address the text clustering (TC). The offered"

(2) There seem to be unnecessary amount of citations to the works performed by one other group. Namely, citations 47,10,11 and then 12 and 13 (where most of these work seem to be highly connected to each other). In order to preserve the integrity of the journal and academic community, I suggest being more selective in your citations.

(3) A disturbing amount of abbreviations and unnecessary nomenclature has been used that is not conductive to the cohesion of the report. For example FA (firefly algorithm) has been used two or three times in the entire document, I would suggest using the complete name "firefly algorithm" rather than FA, in order to not to confuse the reader. Or two different abbreviations TC and TD are used for text clustering and text document clustering. I suspected there is a difference between the two but I couldn't find any distinction between these two, neither in the manuscript, nor online. If these two refer to the same concept, you can use only one of them, if not, you will need to do a better job defining the two in the manuscript.

(4) In the section "population initialization phase" the authors refer to equation (4) for computing cluster centroids. I believe they intended to refer to equation (6).

(5) The authors are using F-value or F-measure as their evaluation metric, I think it is a good practice to include the metric definition in the manuscript. Moreover, when referring to this metric they cite a 2017 paper. You can say that the cited paper has used F-value as a measure of clustering efficiency and therefore it is sound to use F-value, but the way it is put in the manuscript, it implies that the 2017 paper was the “invention” of the F-value which clearly is wrong.

(6) In the section "Parameter setting used in the study": how did the authors come to this setting? Did they perform a parameter study? Or maybe cross-validation?

(7) Number of terms in documents (table 1) seem to be very low. For example 1700 unique words in a dataset of 299 documents (CSTR dataset). Or 3000 words in 900 documents. Maybe this numbers are after stemming and stop-word removal, but how are those steps performed? I would recommend, if not within the main manuscript, at least provide a supplementary material section where you describe this detailed methodology. As an example refer to table1 in the following paper: https://doi.org/10.1109/ACCESS.2019.2923462 Their datasets seem to have much higher term/document ratio, is there a reason for this? Maybe your datasets are more technical than of general nature? in this case your algorithm will be for "technical text classification"

(8) The authors are using tf-idf and vector space model for their document representation. Why not using a more elaborate word-representation such as Word2Vec or GloVe. Is there a benefit in using tf-idf and not more modern representations? Needs to be explained.

I applaud the authors for their efforts but I think the points made above are the minimum that need to be met to before being published in a journal.

6. PLOS authors have the option to publish the peer review history of their article (what does this mean?). If published, this will include your full peer review and any attached files.

Reviewer #1: No

Reviewer #2: No

---

## [Author Response · Author response to Decision Letter 0]

23 Oct 2019

Thank you for the valuable feedback. We have added some sentences/paragraphs in order to response the comments from the reviewers. The following summarizes the updates that have been made to the paper as a response to the reviewers’ comments. Moreover, additional statistical analysis is added in the experimental results section to fulfill the journal requirement that have been addressed by the reviewers. 

Reviewer #1:

1. The motivations for this study need to be made clearer: 

 The motivation is modified in the introduction section, and cleared in the closing paragraph of the related work. 

2. Figures must be in a good quality with high resolution:

 All figures is modified with better resolution according to the journal figures requirements. 

3. The manuscript is poorly written. Authors really need to improve their written communications skills in English: 

 The manuscript grammar is rechecked and proofread.

4. All references must be revised. Issue numbers and volume numbers are missed from most of references: 

 All references mentioned in the comment are revised accordingly. 

5. Conclusions should address whether or not they achieved the objective of the study:

 The conclusion section has been modified to include the achievement of the objective of the study.

Reviewer #2:

1. There are multiple grammatical and editorial errors in the text:

 The manuscript grammar is rechecked and proofread. 

2. There seem to be unnecessary amount of citations to the works performed by one other group. Namely, citations 47,10,11 and then 12 and 13:

 References 11, 12, and 47 is removed. The remaining references are kept since they represent different works in text clustering such as multiobjective optimization, unsupervised feature selection, and Hybrid krill herd algorithm approaches

3. A disturbing amount of abbreviations and unnecessary nomenclature has been used that is not conductive to the cohesion of the report. You will need to do a better job defining the two different abbreviations TC and TD in the manuscript:

 All abbreviations have been revised accordingly. The TD and TC are discussion is modified in the introduction section.

4. In the section "population initialization phase" the authors refer to equation (4) for computing cluster centroids. I believe they intended to refer to equation (6);

 The equation number is corrected to be equation (6) (calculation of cluster centroids)

5. The authors are using F-value or F-measure as their evaluation metric, I think it is a good practice to include the metric definition in the manuscript. Moreover, when referring to this metric they cite a 2017 paper;

The unnecessary citation is removed since its only indicates that this measure have been used in similar studies. The equations and discussion of F-measure are added in the experimental setup section.

6. In the section "Parameter setting used in the study": how did the authors come to this setting? Did they perform a parameter study? Or maybe cross-validation?:

 It is based on an experimental basis and the drawing on previous work from the scientific literature [3]*. This statement is added to the experimental setup section.

7. Number of terms in documents (table 1) seem to be very low. For example 1700 unique words in a dataset of 299 documents (CSTR dataset). Or 3000 words in 900 documents. Maybe this numbers are after stemming and stop-word removal, but how are those steps performed? I would recommend, if not within the main manuscript, at least provide a supplementary material section where you describe this detailed methodology:

 These datasets are standard benchmark text documents datasets that already pre-processed, the description is added in the experimental setup section. The detailed steps of the pre-processing are elaborated in background section (text clustering problem).

8. The authors are using tf-idf and vector space model for their document representation. Why not using a more elaborate word-representation such as Word2Vec or GloVe. Is there a benefit in using tf-idf and not more modern representations? Needs to be explained:

 The following are the reasons to adopt the tf-idf in our algorithm:

 1. The TF/IDF is commonly used by TC algorithms, where the frequent terms will be a good indicator for a certain topic [1] [2]*

 2. The standard text document clustering datasets are represented (pre-processed) by tf-idf term frequency.

 More discussion is added to the Text clustering problem section. And also more discussion is added (as in comment No. 7).

*References.

1. Aggarwal CC, Reddy CK. Data Custering Algorithms and Applications. 1st ed. Taylor & Francis Group, LLC; 2013. 

2. Cui X, Potok TE, Palathingal P. Document clustering using particle swarm optimization. Proc 2005 IEEE Swarm Intell Symp 2005 SIS 2005. 2005; 185–191. doi:10.1109/SIS.2005.1501621

3. Mustafa HMJ, Ayob M, Nazri MZA, Kendall G. An improved adaptive memetic differential evolution optimization algorithms for data clustering problems. PLoS One. 2019;14(5): e0216906. doi:10.1371/journal.pone.0216906

---

## [Decision Letter · Decision Letter 1]

18 Feb 2020

PONE-D-19-20564R1

Solving Text Clustering Problem using a Memetic Differential Evolution Algorithm

PLOS ONE

Dear Mr. Mustafa,

Thank you for submitting your manuscript to PLOS ONE. After careful consideration, we feel that it has merit but does not fully meet PLOS ONE’s publication criteria as it currently stands. Therefore, we invite you to submit a revised version of the manuscript that addresses the points raised during the review process.

Please address all the comments given by the reviewers. One of the reviewer raise their concern where the similarity between this paper with other published paper entitle "An improved adaptive memetic differential evolution optimization algorithms for data clustering problems" under PLoS One as well. Please emphasize on the different between these two papers and highlight your contribution explicitly.

We would appreciate receiving your revised manuscript by Apr 03 2020 11:59PM. To enhance the reproducibility of your results, we recommend that if applicable you deposit your laboratory protocols in protocols.io, where a protocol can be assigned its own identifier (DOI) such that it can be cited independently in the future. For instructions see: http://journals.plos.org/plosone/s/submission-guidelines#loc-laboratory-protocols

We look forward to receiving your revised manuscript.

Kind regards,

Mohd Nadhir Ab Wahab, Ph.D.

Academic Editor

PLOS ONE

Additional Editor Comments (if provided):

Please address all the comments given by the reviewers. One of the reviewer raise their concern where the similarity between this paper with other published paper entitle "An improved adaptive memetic differential evolution optimization algorithms for data clustering problems" under PLoS One as well. Please emphasize on the different between these two papers and highlight your contribution explicitly.

Reviewers' comments:

Reviewer's Responses to Questions

**Comments to the Author**

1. If the authors have adequately addressed your comments raised in a previous round of review and you feel that this manuscript is now acceptable for publication, you may indicate that here to bypass the “Comments to the Author” section, enter your conflict of interest statement in the “Confidential to Editor” section, and submit your "Accept" recommendation.

Reviewer #3: All comments have been addressed

Reviewer #4: (No Response)

Reviewer #5: (No Response)

2. Is the manuscript technically sound, and do the data support the conclusions?

Reviewer #3: Yes

Reviewer #4: Yes

Reviewer #5: No

3. Has the statistical analysis been performed appropriately and rigorously? 

Reviewer #3: Yes

Reviewer #4: Yes

Reviewer #5: No

4. Have the authors made all data underlying the findings in their manuscript fully available?

Reviewer #3: Yes

Reviewer #4: Yes

Reviewer #5: Yes

5. Is the manuscript presented in an intelligible fashion and written in standard English?

Reviewer #3: Yes

Reviewer #4: Yes

Reviewer #5: No

6. Review Comments to the Author

Reviewer #3: The authors have addressed the comments from previous reviewers and improved the manuscript. I suggest all abbreviations of algorithms such as KH, CS, PSO, etc. be removed, they are not necessary and still excessive, making the manuscript hard to follow. The abstract needs to be improved as well.

Reviewer #4: The authors have met all comments of the previous reviewers. The English could still be improved, though.

Reviewer #5: 1. Line 390: in the comparative analysis section: Why is the work from Ref 23 not included here? What is the difference between this work and the work presented in Ref 23?

2. Line 340: The use of F-measure sounds good, but it is logical to present the full spectrum of evaluation metrics before narrowing down to one. For instance, the reader would be interested to know the AUC, in addition to the F-measure.

3. Line 233 to 235: The study failed to explained the limitation of existing algorithm. The use of overly broad terminology does not account for the limitation. Furthermore, the study failed to state the limitation of MA and DE for which an extended is considered important. I think this are missing steps that can improve the paper.

4. Reference to the Related works: The language is really bad. Another round of proof reading would be required. One more thing, the sequence of the reference can be adjusted in ascending order. It is presented in a scattered manner. this does not aid reading.

Overall, the manuscript failed in readability due to poor grammar and sentence structure.

7. PLOS authors have the option to publish the peer review history of their article (what does this mean?). If published, this will include your full peer review and any attached files.

Reviewer #3: Yes: Jijun Tang

Reviewer #4: Yes: Volker Ahlers

Reviewer #5: No

---

## [Author Response · Author response to Decision Letter 1]

2 Apr 2020

Thank you for the valuable feedback. We have added some sentences/paragraphs to respond to the comments from the reviewers. The following table summarizes the updates that have been made to the paper as a response to the reviewers’ comments. Moreover, additional statistical analysis is added in the experimental results section to fulfill the journal requirement that has been addressed by the reviewers. 

Reviewer #3 comment # 1: I suggest all abbreviations of algorithms such as KH, CS, PSO, etc. be removed, they are not necessary and still excessive, making the manuscript hard to follow.

Response to comment :

The abbreviations are used within the related work and the comparison with the state of art sections. We defined these abbreviations whenever are used, then the abbreviations are used due to difficulties of using the full name of the algorithms within the figures and tables, for example, hybrid krill herd algorithm (MMKHA).

Reviewer #3 comment # 2: The abstract needs to be improved as well.

Response to comment :

We revised the abstract and made proofreading.

Reviewer #4 comment # 1: The English could still be improved.

Response to comment :

Proofreading is made.

Reviewer #5 comment # 1: Line 390 (460 in the new manuscript): in the comparative analysis section: Why is the work from Ref 23 not included here? What is the difference between this work and the work presented in Ref 23?

Response to comment :

Ref 23 (21 in the new manuscript) uses different objective functions and local search heuristic. In Ref 23, the datasets used to evaluate the algorithm is low dimension benchmark datasets that are collected from many domains. More discussion about this issue can be found in the related work section (las paragraphs line .. ). The text clustering datasets (as mentioned in the background section) use different kind of data (tf/idf) and requires different objective function, evolutionary steps, and local search to be proposed.

Reviewer #5 comment # 2: Line 340 (403 in the new manuscript): The use of F-measure sounds good, but it is logical to present the full spectrum of evaluation metrics before narrowing down to one. For instance, the reader would be interested to know the AUC, in addition to the F-measure.

Response to comment :

The AUC metric and ROC are included in the study. A discussion about these evaluation metrics is added to the experiment setup section. Besides, the ROC curve (figure 5) and the results of the AUC metric (tables 3-5) are added and compared using the statistical analysis in the experimental results and discussion section. The abstract, introduction, and conclusion are revised accordingly.

Reviewer #5 comment # 3: Line 233 to 235 (235 in the new manuscript): The study failed to explain the limitation of the existing algorithm. The use of overly broad terminology does not account for the limitation. Furthermore, the study failed to state the limitation of MA and DE for which an extended is considered important. I think these are missing steps that can improve the paper

Response to comment :

A paragraph is added before the comment to elaborate the need to hybridize between DE and MA with three benefits (line 235 - 251).

Reviewer #5 comment # 4: Reference to the Related works: The language is really bad. Another round of proofreading would be required. One more thing, the sequence of the reference can be adjusted in ascending order. It is presented in a scattered manner. This does not aid reading. Overall, the manuscript failed in readability due to poor grammar and sentence structure

Response to comment :

We reorganized and revised the related work section accordingly. The sequence of references is adjusted. The related work is grouped according to the metaheuristic algorithm or approach used.

---

## [Decision Letter · Decision Letter 2]

23 Apr 2020

Solving Text Clustering Problem using a Memetic Differential Evolution Algorithm

PONE-D-19-20564R2

Dear Dr. Mustafa,

We are pleased to inform you that your manuscript has been judged scientifically suitable for publication and will be formally accepted for publication once it complies with all outstanding technical requirements.

With kind regards,

Mohd Nadhir Ab Wahab, Ph.D.

Academic Editor

PLOS ONE

Additional Editor Comments (optional):

Congratulations. Please proof read the article as well.

Reviewers' comments:

Reviewer's Responses to Questions

**Comments to the Author**

1. If the authors have adequately addressed your comments raised in a previous round of review and you feel that this manuscript is now acceptable for publication, you may indicate that here to bypass the “Comments to the Author” section, enter your conflict of interest statement in the “Confidential to Editor” section, and submit your "Accept" recommendation.

Reviewer #3: All comments have been addressed

Reviewer #4: All comments have been addressed

2. Is the manuscript technically sound, and do the data support the conclusions?

Reviewer #3: Yes

Reviewer #4: (No Response)

3. Has the statistical analysis been performed appropriately and rigorously? 

Reviewer #3: Yes

Reviewer #4: (No Response)

4. Have the authors made all data underlying the findings in their manuscript fully available?

Reviewer #3: Yes

Reviewer #4: (No Response)

5. Is the manuscript presented in an intelligible fashion and written in standard English?

Reviewer #3: Yes

Reviewer #4: (No Response)

6. Review Comments to the Author

Reviewer #3: The authors have adequately addressed all my concerns, I have no further request. for other updates.

Reviewer #4: (No Response)

7. PLOS authors have the option to publish the peer review history of their article (what does this mean?). If published, this will include your full peer review and any attached files.

Reviewer #3: Yes: Jijun Tang

Reviewer #4: Yes: Volker Ahlers

---

## [Editor Report · Acceptance letter]

26 May 2020

PONE-D-19-20564R2 

Solving Text Clustering Problem using a Memetic Differential Evolution Algorithm 

Dear Dr. Mustafa:

I am pleased to inform you that your manuscript has been deemed suitable for publication in PLOS ONE. Congratulations! Your manuscript is now with our production department. 

With kind regards,

on behalf of

Dr. Mohd Nadhir Ab Wahab 

Academic Editor

PLOS ONE